# Design and Optimization of a Permanent Magnet-Based Spring–Damper System

**Nicolò Gori** [1,*] , **Claudia Simonelli** [1] , **Antonino Musolino** [1,*] , **Rocco Rizzo** [1] , **Efren Díez Jiménez** [2] and **Luca Sani** [1]

1   School of Engineering (DESTEC Department), University of Pisa, 56122 Pisa, Italy;
    claudia.simonelli@phd.unipi.it (C.S.); rocco.rizzo@unipi.it (R.R.); luca.sani@unipi.it (L.S.)
2   Mechanical Engineering Area, Universidad de Alcalá, 28805 Alcalá de Henares, Spain; efren.diez@uah.es
*   Correspondence: nicolo.gori@phd.unipi.it (N.G.); antonino.musolino@unipi.it (A.M.);
    Tel.: +39-050-2217101 (N.G.); +39-050-2217321 (A.M.)

**Abstract:** This paper reports the design and optimization of a permanent magnet-based spring. The aim of the optimization, performed using a particular form of the self-organizing map (SOM) algorithm, was to determine the dimensions of a ring PM-based spring with a force–displacement curve similar to a desired one. For each step in the optimization process, a spring composed of different ring-shaped magnets was analyzed using a semi-analytical model. Its characteristic was compared with the desired one to search for a minimum cost function obtained by subtracting the evaluated and the desired force–displacement curve. The resulting algorithm was efficient in the design of a spring with a desired characteristic. The geometry obtained was used to study an electrodynamic damper based on the exploitation of the interaction between the moving magnet of the spring and a conductive cylinder. A parametric analysis was performed: the damping effect grows when the cylinder thickness increases and decreases with the gap between the cylinder and the magnets. Also, the cylinder thickness needed to reduce to one the number of overshoots in the moving magnet's position decreases with the gap increase. Computations were performed using the research code EN4EM (Electric Network 4 ElectroMagnetics) developed by the authors.

**Keywords:** self-organizing maps; permanent magnet spring; damper; optimization; eddy currents





## 1. Introduction

Magnetic springs obtained by the interactions between permanent magnets (PMs) can substitute conventional (metallic) springs in several applications, especially those characterized by long stroke and high-frequency operations. As known, high natural oscillation frequencies require that the mover mass (including a portion of the mass of the spring) is kept as low as possible, but longer strokes imply larger and heavier springs. Metallic springs are usually characterized by material consumption and plastic deformation resulting in poor stability of the system and in debris production. In magnetic springs, these drawbacks are strongly reduced or even absent, making magnetic springs excellent candidates for those applications where cleanliness requirements are essential, such as in aerospace and surgical applications or equipment used in clean rooms for the manufacture of semiconductors.

The availability of high-performance neodymium iron boron (NdFeB) PMs characterized by high remanence and coercivity allows us to achieve ratios of the magnetic force to the weight of the magnetic rings exceeding 100 [1]. This has led to a wide variety of applications in which a high stiffness is required, such as vibration insulators where pneumatic springs are commonly used; the employment of PM-based springs facilitates the elimination of vibrations caused by the contact between the components [2,3]. Other application examples are energy harvesting systems [4] and also those in which the spring is an auxiliary element [5], or bearings [6] for which the contactless nature of a PM-based

spring increases the reliability and the efficiency of the system. These aspects, added to the absence of material fatigue and friction connected to the use of magnetic springs instead of mechanical ones, make these systems particularly suitable for oscillatory actuators [7]. Moreover, PM-based springs are low-noise devices that work without the use of oil, aspects particularly important for aerospace applications [8].

Analytical and semi-analytical models have been developed to investigate different arrangements and shapes of permanent magnet springs (especially in the presence of multiple coaxial disks [9,10] and cylinders [11] with various magnetization directions), and experimental techniques have been developed to identify the force–displacement characteristic of a passive suspension [12]. Analytical and semi-analytical formulations are of quick evaluation, and this makes them well suited for optimization processes where a large number of evaluations of cost functions, which require the calculation of the forces between the magnets, are needed.

The algorithm used in this paper aims to determine the arrangement of PMs to obtain a desired force–displacement curve. A semi-analytical model able to derive the force–displacement characteristic of a given configuration of the system was developed starting from the model described in [9]. The curve given by the model was then compared with the desired force–displacement characteristic. Their difference established the cost function of the optimization process, for which the geometrical parameters of the magnets composing the system are the parameters through which we search for the optimum. The optimization process was based on the self-organizing map (SOM) algorithm described in [13] and used in [14–16]. With a contained computational time, it was able to give consistent results validated with numerical software.

Similarly to all magnetic suspension devices, magnetic springs are characterized by extremely low damping. There are applications (e.g., mechanical resonant energy harvesters, as in [17,18]) in which this characteristic is beneficial to the efficient operation of the device. Other applications, instead, need additional damping in order to mitigate the vibrations occurring during the normal operations (e.g., automobile suspensions [19], rotor dynamics [20], and aerospace [21,22]).

Damping can be obtained by adding to the magnetic springs conventional dampers, like friction or viscous ones, which have excellent dynamical performance. In particular, viscous dampers based on magnetorheological fluids [23–25] featuring semi-active damping represent the state of the art. However, conventional dampers are characterized by debris or oil leakage, thus removing the advantages that make the magnetic springs particularly suitable for clean applications.

A damper can also be obtained by exploiting the drag force caused by the eddy currents induced in a conductor by magnets moving in its proximity. These devices are inherently free of friction and represent a clean and efficient solution, even though the typical damping densities are not so high as in viscous dampers. Research has focused its attention on techniques to improve the damping density. Authors in [26,27] proposed the use of linear magnetic gears to obtain a very high damping density of more than 8 $\text{MNsm}^{-4}$, while [28] described an eddy current damper based on the Halbach array, whose characteristics are very close to those of conventional dampers in automobile suspensions.

In this paper, the authors will focus on the description of a general methodology for (a) the evaluation of the repulsive suspension force of the magnetic spring and (b) the analysis of the damping actions consequent to the movement of a magnet inside a conductive cylinder. The first point is necessary for the design and optimization of the PM-based spring, while the second one also facilitates the dynamic description of the whole system in transient conditions to extract the dynamic parameters of the spring–damper system. The key features of the methodology are described by considering a simple magnetic spring consisting of two repelling annular permanent magnets with damping produced by the eddy currents induced by the upper magnet (the moving one) in a cylindrical conductor.

The paper is organized as follows: Section 2 introduces the geometry of the device and shortly describes the EN4EM code used for the evaluation of the suspension force and

for the electro-mechanical analysis. Section 3 describes the optimization procedure of the magnetic spring, while Section 4 analyzes the damping effects. Final remarks and ideas for future research are reported in the Section 5.

## 2. Modeling

### 2.1. Analyzed Geometry

The model analyzed and optimized in this paper refers to ring-shaped neodymium ($Nd_2Fe_{14}B$) PM-based springs. It is able to deal with systems with a different number of permanent magnets, generalizing the geometry visible in Figure 1, in which a system with two or three ring-shaped magnets is visible. The geometries analyzed are axial-symmetric, but all the following considerations can be applied to more general full 3D shapes even though the computational time will increase since the research code EN4EM was originally developed for 3D geometries and then modified to take advantage of the symmetries.

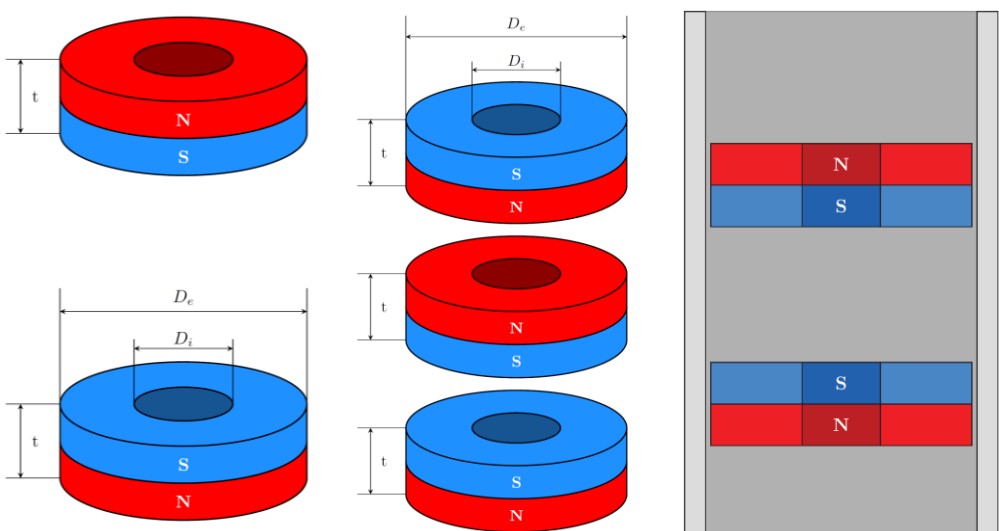

**Figure 1.** PM-based spring composed of two and three ring-shaped magnets with and without conductive damping tube.

The parameters that define the annular geometry are the inner and outer diameters, Di and De, respectively, and the thickness of the disk, t. These parameters are the tunable ones in the optimization process. We assume that both rings are uniformly magnetized along the axial direction with opposite polarities in order to obtain a repulsive force between them. It is known that in this arrangement of PMs, the repulsive force monotonically decreases as the clearance between the magnets increases and assumes its maximum when the clearance is zero. A consequence of the Earnshow theorem is the absence of stable equilibrium points for arrangements of PMs and stationary (constant) currents only. This means that a containment has to be introduced to maintain the coaxial configuration. The inner part of the rings can be used to accommodate a cylindric rod that ensures a linear guiding of the moving magnet while preventing rotations and transverse translations. When the rod is conductive, as in [18], the vertical movement of the upper PM ring induces eddy currents in it that contribute to damping. In the present study, we consider the damping effects of an external coaxial cylindric conductor as in the rightmost geometry shown in Figure 1.

### 2.2. Model Development and Implementation

The optimization of a ring geometry usually requires the evaluation of a great number of cost functions built by evaluating the force profile as a function of the clearance between the PMs. When the rings are made of high-quality NdFeB material, and the magnetization can be assumed as uniform, it is possible to substitute the material with an arrangement of equivalent currents distributed on the surfaces of the rings and consider the region occupied

by the magnets as a vacuum. In the case of uniform axial magnetization, the equivalent currents are distributed in the inner and outer lateral cylindric rings' surfaces (see Figure 2). Considering the hypothesis $B_r \cong \mu_0|H_c|$, usually assumed for NdFeB PMs, the equivalent magnetization currents $\boldsymbol{J}_{eq} = \boldsymbol{n} \times \boldsymbol{M}$ for an axially magnetized ring ($\boldsymbol{M} = M_z > 0$) are $J_{\vartheta,i} = \boldsymbol{n}_{r,i} \times \boldsymbol{M} = -|H_c|$ $A/m$ and $J_{\vartheta,e} = \boldsymbol{n}_{r,e} \times \boldsymbol{M} = -|H_c|$ $A/m$, and they flow in the inner and outer cylindrical lateral surface, respectively. These currents entirely account for the presence of the magnet.

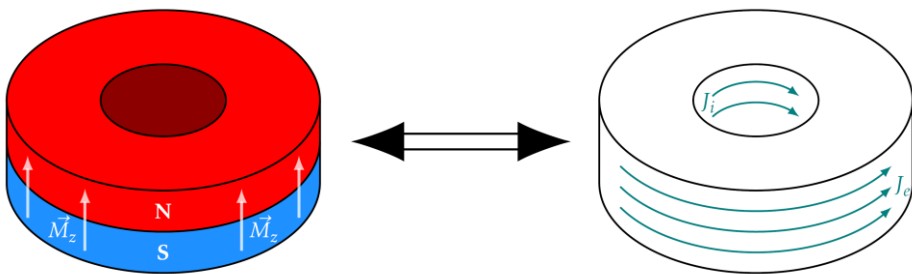

**Figure 2.** Equivalent magnetization currents.

Once the magnets are substituted by their equivalent currents, it is possible to use the model proposed in [9] and embedded in the EN4EM code to efficiently calculate the force–displacement curve for a system of two (or more) ring-shaped magnets. The model implemented in EN4EM requires a reduced number of numerical integrations, making the optimization process to obtain the desired force–displacement characteristic relatively fast.

The evaluation of the force–displacement characteristic of a ring-shaped PM-based spring requires the solution of a magnetostatic problem to determine the interaction between the magnetic fields produced by the PMs when they are at a certain distance. The analysis of the damping is a typical eddy current problem whose solution is based on the diffusion equation of the magnetic field. The eddy currents are induced by the moving magnet, and their intensity depends on the magnet's speed, strictly relating to the force determined through the solution of the magnetostatic problem. In this specific case, we have to solve a strongly coupled electro-mechanical load-driven problem since electrical quantities (i.e., the induced currents) determine the damping force, which enters in the evaluation of the speed of the moving upper ring, which determines the motional electromagnetic force.

The coupled electro-mechanical analysis is a challenging and still open problem characterized by several critical issues. Most commercial codes based on the Finite Element Method (FEM) provide packages for accurate coupled electro-mechanical analyses, but they are typically characterized by long computation times.

Integral Formulations (e.g., the Method of Moments) are more efficient since they require the discretization of active regions only (conductors and PMs in this case), and they do not need to couple meshes of components with different speeds, avoiding the time consumed for this task. It is worth pointing out how the Integral Formulations can produce accurate results by using coarse discretization (when compared with those required by FEM).

The numerical investigations about the damping effects in the magnetic springs were carried out using the code "EN4EM" (Equivalent Network 4 Electromagnetic Modeling). It is based on an Integral Formulation that transforms the diffusion equations of the induced eddy currents into the governing equations of an equivalent electric network, whose parameters depend on the geometry (i.e., the relative positions) of the active elements (conductors and PMs) of the system [29,30].

Figure 3 shows the steps of a conceptual flow chart of the numerical formulation:

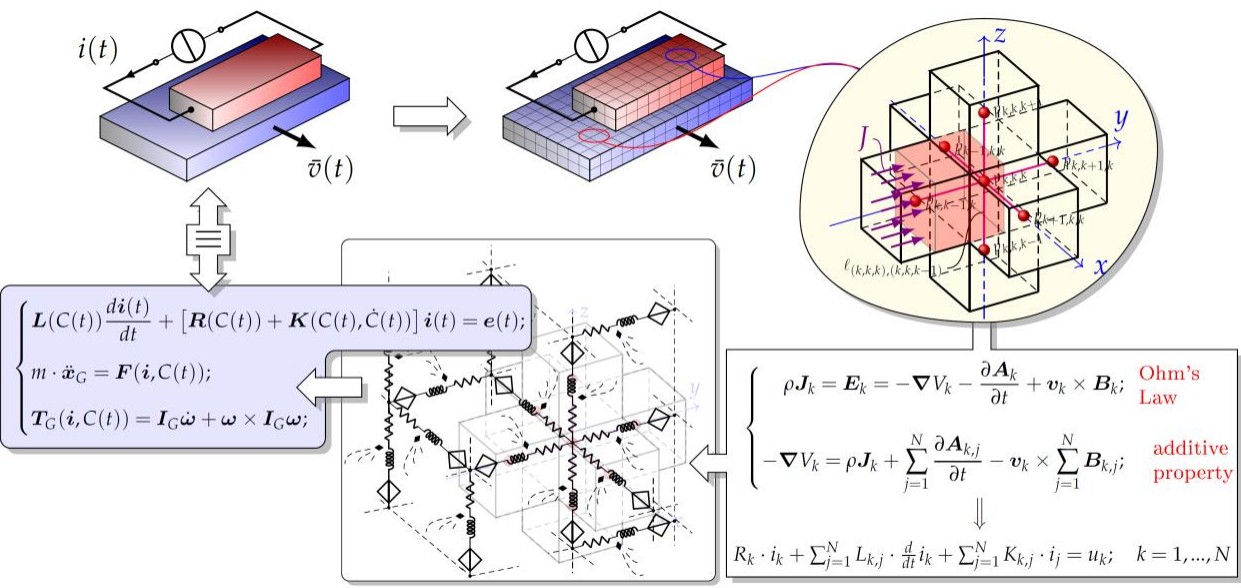

**Figure 3.** Conceptual flow chart of the numerical formulation. Reprinted with permission from Ref. [31]. 2017, IEEE.

(a) Discretization in elementary volumes: slabs and ring sectors with a rectangular cross-section are chosen because of the availability of efficient formulas for the evaluation of fields and potentials [32,33];

(b) Connecting the centers of near elements (red dots in the upper-right corner drawing of Figure 3) to form a 3D-structured grid (red segments);

(c) Associate with each segment of the grid a new elementary volume highlighted in light red (upper-right corner of Figure 3); the current density in each new volume has the same direction as the segment; intersections of new elementary volumes (up to three) mean that the current density has more than one component;

(d) Writing of Ohm's law: the pointwise version of Ohm's law is written in every new elementary volume of the conductive regions, and it is integrated along the current direction and averaged on the cross-section;

(e) Arrangement in an electric network: as a result of point (b), a set of equations is obtained that can be seen as the voltage–current equation of a branch that is a series connection of a resistor, an inductor coupled with other inductors, and a voltage generator controlled by the currents in other elementary volumes;

(f) Writing of the governing electro-mechanical equations: mesh analysis is adopted to write the governing equation of the electric network, while Newton's law is written for the mechanical analysis.

The electric governing equations of the coupled electro-mechanical system can be written as:

$$L(\boldsymbol{\xi})\frac{d}{dt}\boldsymbol{i} + \left[\boldsymbol{R}(\boldsymbol{\xi}) + \boldsymbol{K}\left(\boldsymbol{\xi},\dot{\boldsymbol{\xi}}\right)\right]\boldsymbol{i} = \boldsymbol{u}(t), \tag{1}$$

where $\boldsymbol{\xi}$ is a vector including the set of positions and orientations of the elementary volumes at the instant $t$, and $\dot{\boldsymbol{\xi}}$ is its derivative. In the case of systems with only one translating element characterized by one degree of freedom, the term $\boldsymbol{K}\left(\boldsymbol{\xi},\dot{\boldsymbol{\xi}}\right)$ accounting for the motional emf can be written as $v_z \boldsymbol{H}(\boldsymbol{\xi})$. The vector of the currents $\boldsymbol{i}$ also includes the equivalent magnetization currents.

Newton's law for the translational motion along the $z$ direction is written as:

$$F_z(\boldsymbol{\xi}) = m\dot{v}_z, \tag{2}$$

where $F_z$ is the resultant of the forces acting on the moving ring, also including the repulsive force between the ring-shaped PMs and the force due to the eddy currents, both written as Lorentz forces. $F_z$ also contains other forces such as friction, weight, and, more in general, external driving forces. The other terms of Equation (2) represent the mass of the moving body ($m$), and the time derivative of its velocity along the $z$ direction ($\dot{v}_z$), while $\boldsymbol{\xi}$ is the same vector as presented in Equation (1). The general expression of Lorentz force exerted by the $j$-th domain on the $k$-th one is written as:

$$f_{j,k} = \int_{\Gamma_k} \boldsymbol{j}_k \times \boldsymbol{B}_{k,j} d\Gamma \tag{3}$$

where $\boldsymbol{j}_k$ is the current density in $\Gamma_k$ (the k-th elementary domain), and $\boldsymbol{B}_{k,j}$ is the magnetic flux density produced in $\Gamma_k$ by the current in $\Gamma_j$. If we consider the domains relating to the equivalent magnetization currents of the PMs, we can evaluate the repulsive force between them:

$$F_{2,1} = \int_{\Gamma_{2i}} \boldsymbol{j}_{2i} \times \boldsymbol{B}_{2i,1i} d\Gamma + \int_{\Gamma_{2i}} \boldsymbol{j}_{2i} \times \boldsymbol{B}_{2i,1e} d\Gamma + \int_{\Gamma_{2e}} \boldsymbol{j}_{2e} \times \boldsymbol{B}_{2e,1i} d\Gamma + \int_{\Gamma_{2e}} \boldsymbol{j}_{2,e} \times \boldsymbol{B}_{2e,2e} d\Gamma \tag{4}$$

In this equation $\boldsymbol{j}_{2i}$ is the equivalent magnetization current flowing in the inner cylindrical lateral surface of the second magnetized ring $\Gamma_{2i}$ (the upper one), $\boldsymbol{B}_{2i,1i}$ is the magnetic flux density produced in $\Gamma_{2i}$ by the magnetization current $\boldsymbol{j}_{1i}$ flowing in $\Gamma_{1i}$.

The damping force is similarly evaluated integrating on $\Gamma_{2i}$ and $\Gamma_{2e}$ the magnetic flux density produced by currents induced by the motional emf on the conductive cylinder.

When Equation (1) is written for the PMs of the magnetic spring, the voltage generators on the right-hand side of the equations are zero, while the equivalent magnetization currents are constant. The forcing terms in the equations are the motional emf due to the relative movement of the upper magnet with respect to the conductive cylinder. Equation (1) can be rewritten as:

$$\widetilde{L}(\boldsymbol{\xi})\frac{d}{dt}\boldsymbol{i}' + \widetilde{R}(\boldsymbol{\xi})\boldsymbol{i}' = -v_z\boldsymbol{H}(\boldsymbol{\xi})\boldsymbol{J}_{eq}, \tag{5}$$

where $\boldsymbol{J}_{eq}$ represents the equivalent currents of the moving PM and $\boldsymbol{i}'$ the eddy currents in the cylinder. Because of the presence of $v_z$, the currents and the forces depend on the relative speed. Moreover, because of the Faraday–Lenz law, the eddy currents have the effect of contrasting the cause that produces them, in this case, the motion of the upper PM.

EN4EM is a research code that is under continuous development by the authors for investigating electro-mechanical systems [34–37].

*2.3. Validation of the Model*

The model of the spring–damper system was implemented in the EN4EM framework and validated by comparison with the results produced by a commercial FEM code [38]. We determined the force–displacement curve of a PM-based spring with two and three neodymium magnets. Then, we considered the motion of a two-PM spring with and without eddy currents damping.

Two and three identical rings ($r_i$ = 13 mm, $r_e$ = 20 mm, $h$ = 6 mm) uniformly magnetized in the axial direction ($B_r$ = 1.3 T and $H_c$ = 955 kA/m) are used for the construction of the analyzed springs, as shown in Figure 4, while Figure 5 shows the force–displacement characteristics of the two springs.

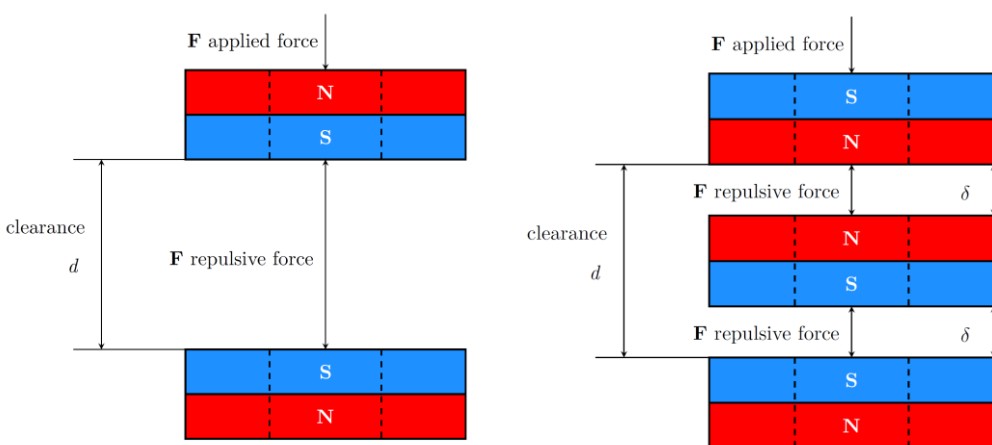

**Figure 4.** Schematic view of the two and three rings magnetic spring.

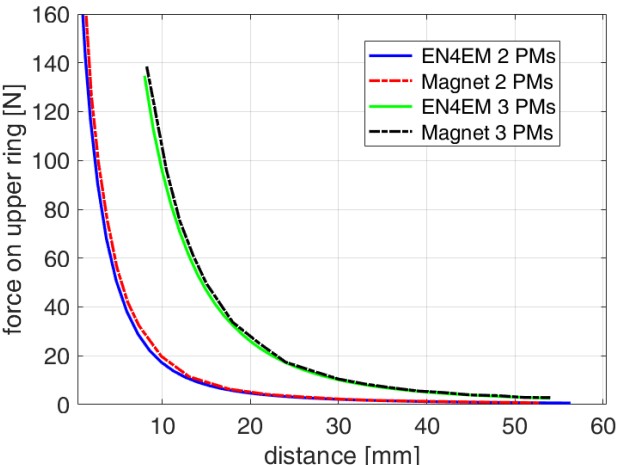

**Figure 5.** Force–displacement characteristic for the two- and three-PM magnetic springs.

Figure 5 reports the simulation results obtained with the commercial FEM package MagNet and EN4EM in terms of force–displacement characteristics. The clearance between the top and the bottom magnet is on the horizontal axis. The minimum clearance between PMs is set to 1 mm, resulting, in the case of a three-PM spring, in a minimum distance between the upper magnet and the lower one of 8 mm. The agreement between the results of the two software packages is good. Because of the use of analytical expressions implemented in EN4EM, it was about twice as fast as MagNet.

The force–displacement characteristic is markedly nonlinear. The spring stiffness increases when the distance between the magnets decreases. We can also observe that the characteristic reduces its nonlinearity when the number of magnets grows, as shown by the presence of a less pronounced knee in the three-PM spring.

It is worth observing that in the three-PM spring, the central magnet floats between the others. Accidental contacts between the magnets cannot be excluded in case of perturbations at the resonance frequency of the central magnet's oscillation. Suitable solutions must be used to ensure an equal distance between the central and external magnets to avoid the damage of PMs that are notoriously brittle. For this reason, we considered only two-PM springs in the rest of the paper.

We also considered the dynamic behavior of the two-magnet spring with and without the eddy currents damping. A mass of 1 kg was added to the upper magnet, and the clearance between the two magnets was set to 1 mm. This was the initial configuration of the system. The damper consists of an aluminum hollow cylinder with inner and outer radii $r_{i,Al} = 20.5$ mm and $r_{e,Al} = 25.5$ mm, respectively, and height $h_{Al} = 10$ cm.

The dynamic of the system without damping is characterized by persistent oscillations characterized by a peak amplitude of about 70 mm, as shown in Figure 6. The force–displacement characteristic (ref. Figure 5) is entirely spanned during the motion. A linear approximation of the characteristic is not valid as it is shown by the non-sinusoidal waveforms shown in Figures 6 and 7, where the comparisons with the results obtained from the FEM package MagNet are reported.

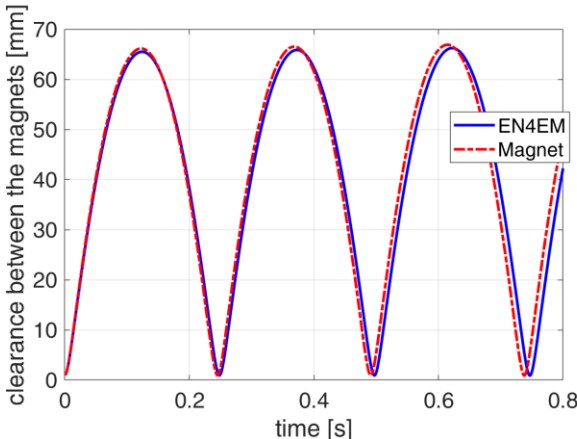

**Figure 6.** Displacement of the upper magnet in the two-PM magnetic spring without damping.

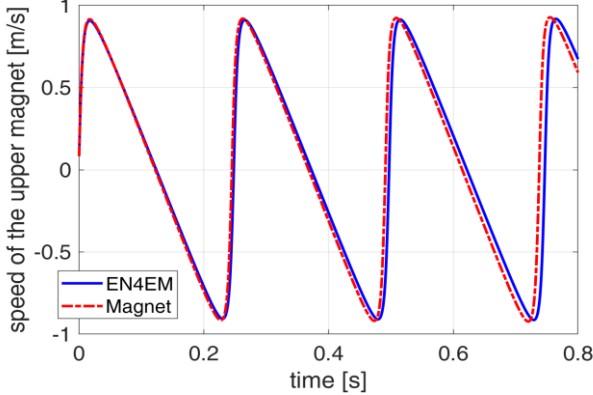

**Figure 7.** Speed of the upper magnet in the two-PM magnetic spring without damping.

To complete the validation, we finally show in Figures 8–10 the comparison of the dynamic behavior of the damped system obtained using EN4EM and MagNet. Also in this case, the agreement between the results is very good.

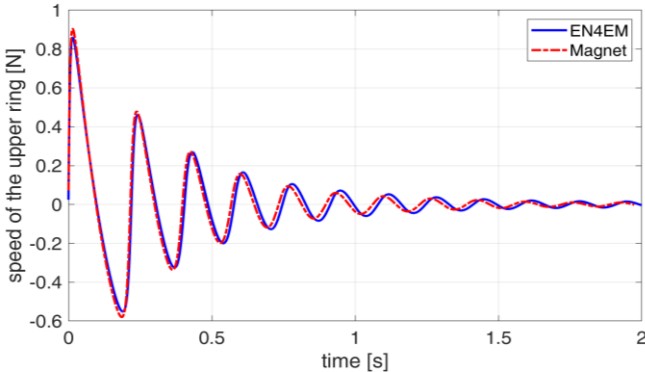

**Figure 8.** Speed of the upper magnet in the two-PM magnetic spring with damping.

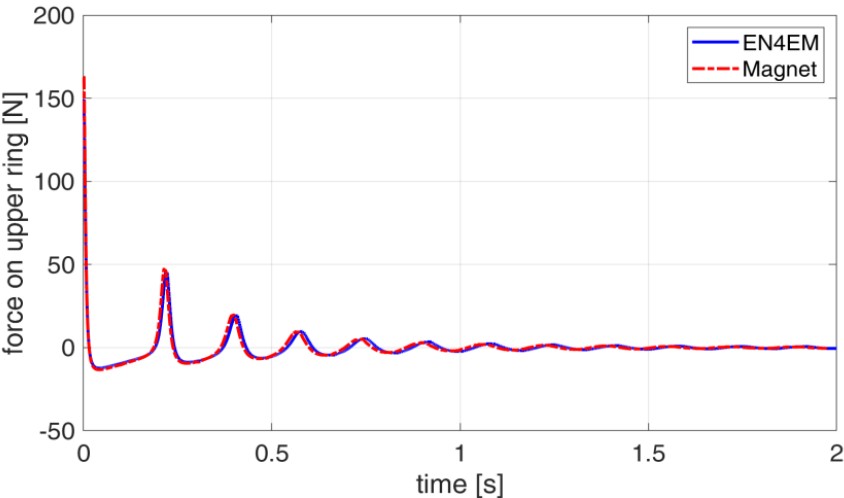

**Figure 9.** Resultant force on the upper magnet in the two-PM magnetic spring with damping.

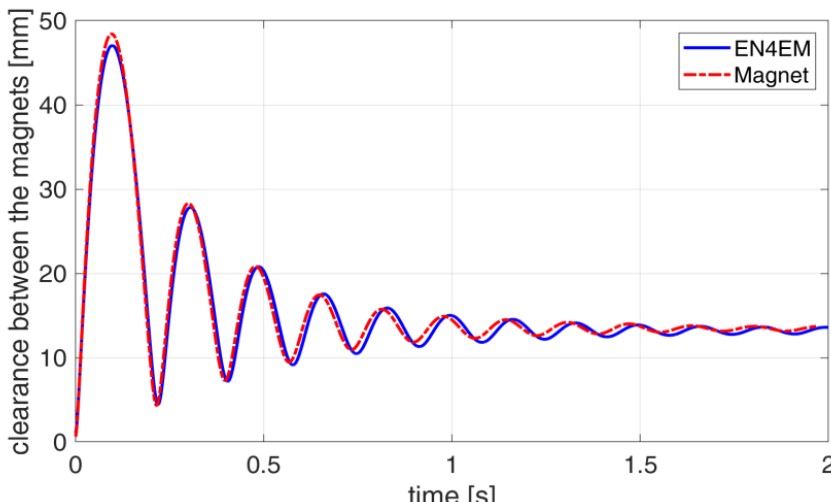

**Figure 10.** Displacement of the upper magnet in the two-PM magnetic spring with damping.

It is interesting to observe how the plotted quantities tend to assume sinusoidal waveforms as the peak-to-peak displacement reduces. For small displacements, it is then possible to use a linearized model.

## 3. Optimization Procedure

One of the main activities of the design of a magnetic spring is the achievement of a prescribed force–displacement characteristic. As observed, the response of a magnetic spring is strongly nonlinear. In comparison with a "mechanical" spring whose behavior can be considered linear, and whose characteristic is completely described by two points, the design of a magnetic spring is more difficult. On the other hand, the intrinsic nonlinearity opens up applications that would otherwise not be feasible with linear systems.

Except for special cases, the force between two magnets cannot be expressed by closed formulas in terms of the geometry of the system. This does not allow the direct application of analytical methods to impose the force values in correspondence with assigned displacements or to minimize (in a suitable way) the distance between the required force–displacement characteristic and that of the generic pair of magnets.

Determining the geometry of the magnets requires more refined optimization procedure.

### 3.1. Cost Function

Using the presented model, it is possible to determine the force–displacement characteristic for a system composed of a given number of ring magnets with fixed dimensions, indicated with $Fc(d, N, g)$, where $d$ is the displacement, $N$ is the number of magnets used for the spring, and g is a vector containing the dimensions of each magnet. The material of the PMs to be used in the design is assigned. In this case, we assume an N42 NdFeB magnet with a remanence $B_r = 1.3$ T and a coercivity $H_c \cong 955$ kA/m. For a ring-shaped magnet, $g$ is a vector with three components representing the three variables that define a ring ($Di$, $De$, and $t$). If the number of ring magnets is $N$, $g$ has $3N$ components, which are the object of the optimization process. $Fc(d, N, g)$ was compared with a desired force–displacement curve $F(d)$ to identify the arrangement of the permanent magnet spring that minimizes the error between the desired and the owned force–displacement characteristic. Once $F(d)$ was defined, the number of magnets was fixed, and the cost function was written as:

$$\text{MSE(g)} = \frac{1}{K}\sum_{j=1}^{K}\left|F(d_j) - F^c(d_j, \hat{N}, g)\right|^2,\tag{6}$$

where $\hat{N}$ is the fixed number of magnets, and $d_j$, $j = 1 \ldots K$, are the equally spaced points inside a given displacement interval in which the force is evaluated.

To better define the optimization problem, constraints on the size of magnets are set: in particular, for each geometrical parameter (i.e., $D_i$, $D_e$ and $t$), a superior and inferior limit is set. The optimization process consists of solving the problem described by the following equations:

$$\begin{cases} g_{opt} = \min\left(MSE\left(g\right)\right) \\ \quad g_{opt} \in R^{sol} \end{cases},\tag{7}$$

in which $R^{sol}$ is the space of solutions, that is, the subspace of $R^{3\hat{N}}$ in which the set constraints are respected.

### 3.2. Optimization Technique

The optimization process has the aim of finding a global minimum of the function in (6) that has several local minima. To solve the problem listed in (7), an evolutionary optimization algorithm is required. Among all the evolutionary algorithms (e.g., particle swarm optimization and genetic algorithms), we used the one described in [13] that has been revealed to be highly effective [14–16].

It is a population-based algorithm based on self-organizing maps structured as follows:

- Initialization of the SOM: definition of a set of centroids belonging to the space of solutions;
- Evaluation of the cost function in all of the defined centroids;
- Evaluation of the fitness function: the winner centroid is the one with the minimum associated fitness function;
- Perturbation of all the centroids: the values of the centroids are perturbated to find any local minima of the associated fitness function;
- The process restarts by updating the values of the centroids considering the winner centroid found in the last step, and the local minima observed perturbating the previous centroids.

### 3.3. Optimized Result

The described algorithm was used to optimize a PM-based spring able to have a force–displacement characteristic, *F(d)*, with an optimal fit with the linear characteristic in the displacement range $5 \div 15$ mm, as reported in Figure 11. The obtained magnetic spring will have a characteristic similar to a mechanical spring ($F_z = -kd$), at least in the selected range.

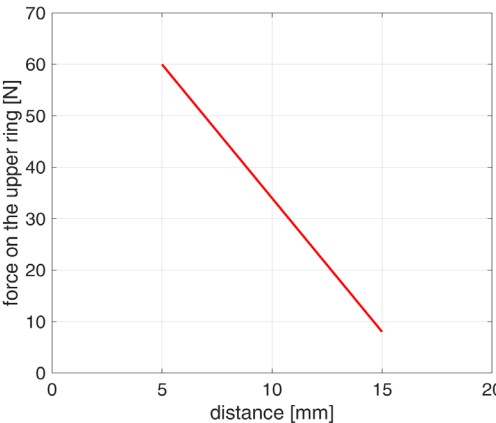

**Figure 11.** Desired force–displacement characteristic.

Comparing the desired curve and the curves reported in Figure 5, it was possible to determine the number of magnets in the spring, which gave a force–displacement characteristic similar to the desired one. In this case, the relatively small required minimum distance (5 mm) prevents the use of an intermediate magnet of reasonable thickness; we have chosen $\hat{N} = 2$, and the optimization process has six geometrical parameters as output. To determine them, 100 displacement points ($K = 100$) were considered using a $10 \times 10$ grid for the SOM. The optimization concluded when one of the following conditions was respected:

- The global optimum improves by a quantity smaller than a given tolerance for a fixed number of iterations;
- The number of iterations performed overtakes a maximum value.

As visible from Figure 12, the first condition determined the convergence of the optimization process. This result was obtained after a computation time of 240 s using a six-core Intel CPU running a MATLAB script.

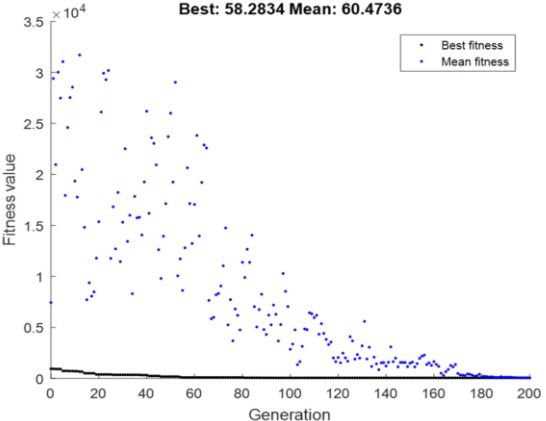

**Figure 12.** Convergence of the optimization algorithm.

The optimized ring-shaped magnets have an internal diameter of 32 mm, an external one of 52 mm, and a thickness of 5 mm. Figure 13 reports the comparison between the characteristic force–displacement of the designed magnetic spring and the desired force characteristic. The similitude of the two curves in the range of the considered displacement is evident.

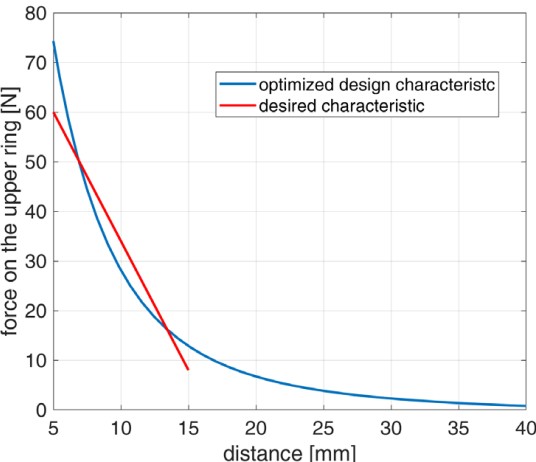

**Figure 13.** Comparison between the desired force–displacement curve and the one associated with the optimized design of the system.

## 4. Eddy Current Damping

Once the optimized design of the magnetic spring has been obtained, we carried out an exhaustive analysis of the damping characteristics of a simple conductive cylindrical surface coaxial with the moving magnet (the upper one) (see Figure 1). We performed several simulations for different values of the clearance between the moving magnet and the conductive surface and of its thickness.

Considering a proper coating of the PMs to avoid any contact between the magnet and the damper, a minimum gap between the magnet and the cylinder of 0.5 mm was assumed. The other considered values of the gap are 1.0 mm and 1.5 mm. The damper was made of aluminum ($\rho_{Al} = 2.86 \times 10^{-8}$ $\Omega$m), and its thickness varies from 1.0 mm to 7.0 mm.

With regard to the initial configuration, we assumed the one where the spring is compressed (initial distance between the PMs equal to 3.0 mm) by a load that is suddenly removed. We considered the natural response of the system with an additional mass $m_{add} = 250$ gr added to that of the upper magnet ($m_{mag} \cong 50$ gr).

Figures 14–19 report the speed and the vertical position of the upper magnet as a function of time for the three different gaps and with various values of the cylinder thickness.

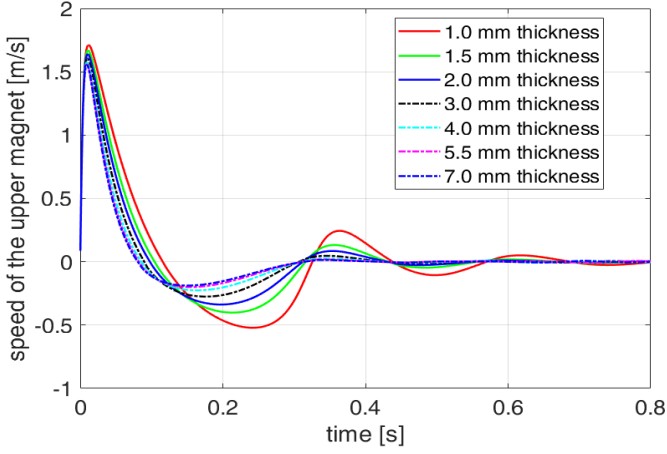

**Figure 14.** Speed of the upper magnet with a gap of 0.5 mm and for different values of the damper thickness in the range 1 ÷ 7 mm.

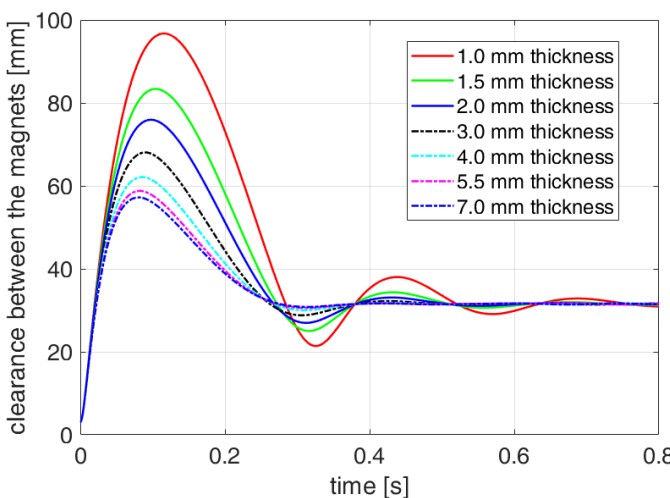

**Figure 15.** Distance between the magnets with a gap of 0.5 mm and for different values of the damper thickness in the range $1 \div 7$ mm.

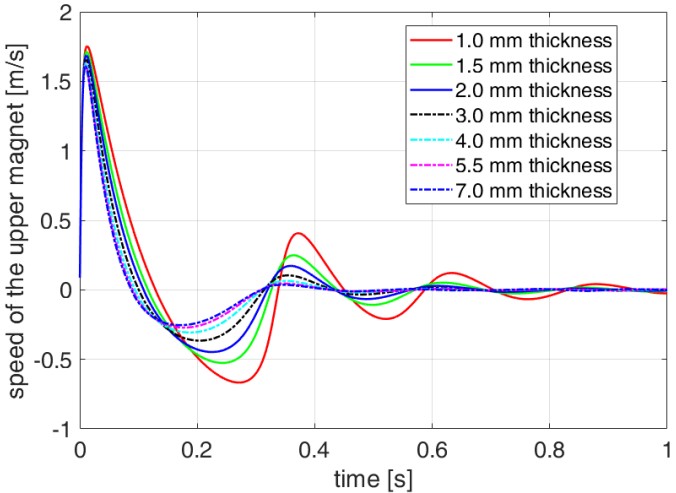

**Figure 16.** Speed of the upper magnet with a gap of 1.0 mm and for different values of the damper thickness in the range $1 \div 7$ mm.

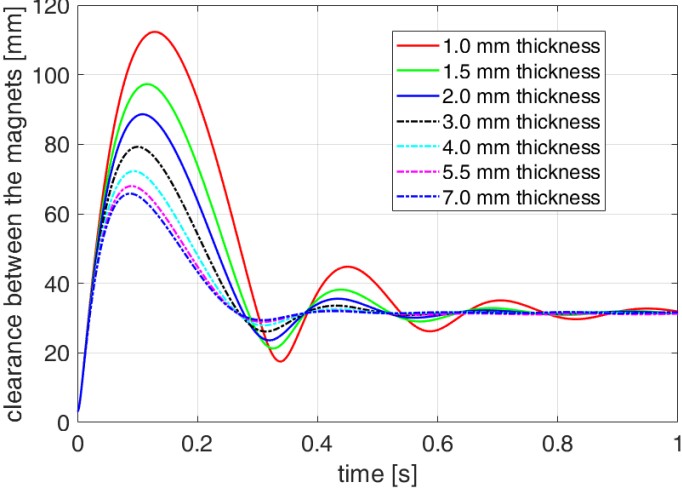

**Figure 17.** Distance between the magnets with a gap of 1.0 mm and for different values of the damper thickness in the range $1 \div 7$ mm.

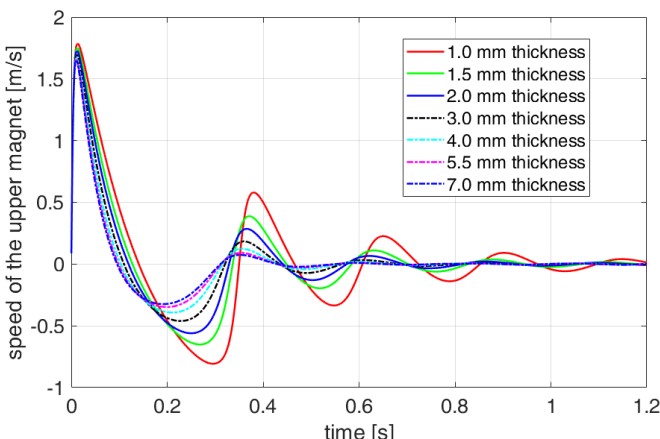

**Figure 18.** Speed of the upper magnet with a gap of 1.5 mm and for different values of the damper thickness in the range 1 ÷ 7 mm.

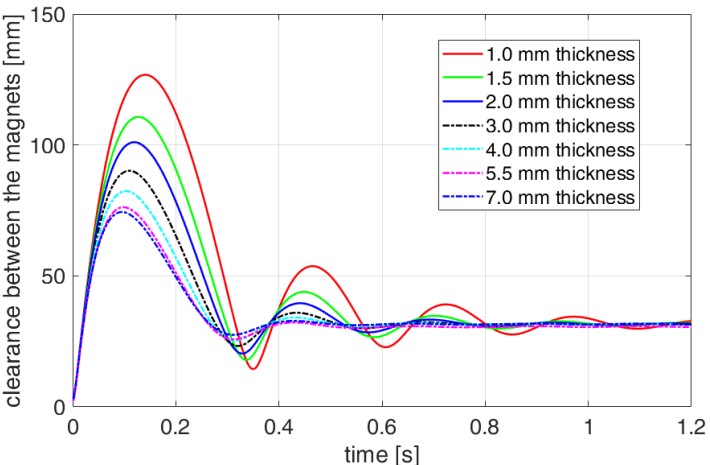

**Figure 19.** Distance between the magnets with a gap of 1.5 mm and for different values of the damper thickness in the range 1 ÷ 7 mm.

Figures 14 and 15 refer to the minimum gap of 0.5 mm, Figures 16 and 17 refer to the gap of 1.0 mm, and Figures 18 and 19 refer to the maximum gap of 1.5 mm. As expected, the damping effect increases with the thickness of the conductive cylinder and decreases with the gap between the cylinder and the moving magnet. A further increase in the conductor thickness does not produce appreciable improvement in the damping; the thickness of 7.0 mm can be considered the one that maximizes the damping for the analyzed system. Moreover, choosing a thickness smaller than 1.0 mm reduces the damping effect too much, especially for a gap of 1.5 mm, making the damper inefficient.

Figure 15 shows that with a gap of 0.5 mm, a conductor with a thickness of 3.0 mm is enough to avoid further appreciable overshoots, except for the first. The thickness has to be increased to 4.0 mm when the gap is 1.0 mm (ref. Figure 17) and to 6.0 mm when the gap is 1.5 mm (ref. Figure 19).

To complete the analysis of the damping effects, we considered the dynamic behavior of the system with three different loads on the upper magnet, a gap of 0.5 mm, and a conductor thickness of 3.0 mm. Figure 20 shows the waveforms of the distance between the magnets with the total masses 300 gr, 350 gr, 450 gr, and 550 gr. As expected, the higher the mass, the higher the natural frequency of the system, while the damping ratio decreases with increasing mass. The equilibrium positions are obviously lower for an increasing mass.

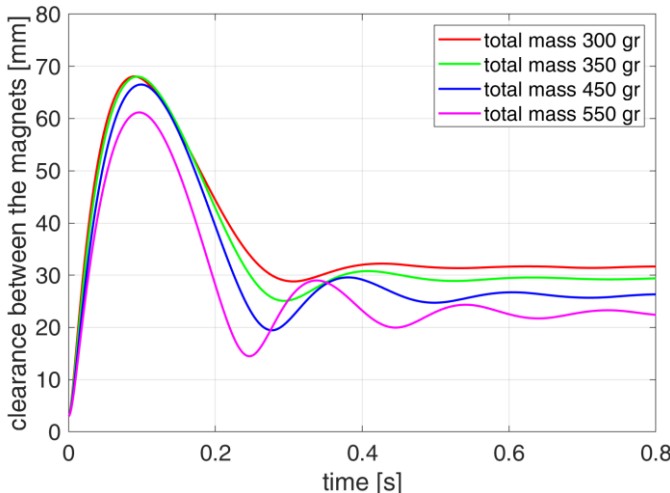

**Figure 20.** Distance between the magnets with a gap of 0.5 mm and damper thickness of 3 mm and different total masses.

As a last remark about the damping effects of the eddy currents, we also considered the effect of a hollow conductive cylinder in the inner part of the PM rings that can be used as a support and a guide for the moving PM [21].

For the two-PM spring considered above (same geometry, total mass, and initial conditions), we have compared the damping effect given by the inner cylinder alone (gap with respect to the PM equal to 0.5 mm, radial thickness 5.0 mm), the outer cylinder alone (gap with respect to the PM equal to 0.5 mm, radial thickness 5.0 mm) and both. Simulation results are in Figure 21, which reports the waveforms of the distance between the magnets. As expected, the damping effect of the outer conductor is stronger than that of the inner one, and the combined action is even stronger.

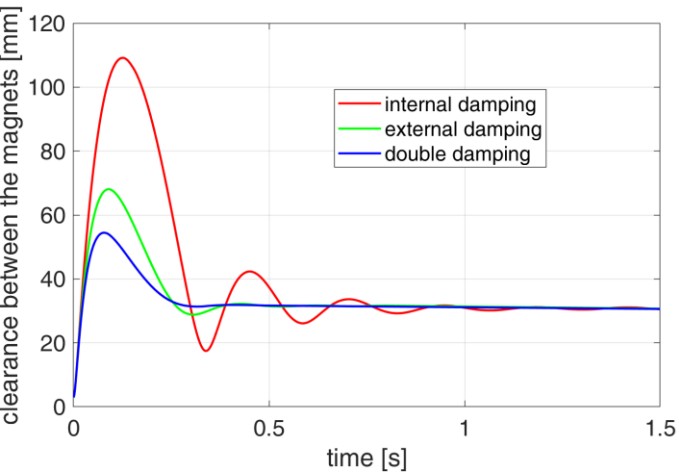

**Figure 21.** Distance between the magnets with different dampers.

## 5. Conclusions

In this paper, a design and optimization technique for magnetic springs was introduced. Starting from a desired force–displacement characteristic, the algorithm is able to optimize the geometry of a PM-based spring to have the requested curve. The procedure is based on a semi-analytical model implemented in the research code EN4EM, which can also consider more general geometries of the magnets. The model was validated by comparison with results obtained from commercial FEM packages. The optimization process has been proven to converge with a limited computational effort. In particular, the optimization of a two-magnet spring to achieve a linear characteristic took 240 s to converge using a

six-core Intel CPU running a MATLAB script. The configuration so determined is used to perform with EN4EM a parametric analysis of a system exploiting the electrodynamic damping provided by the eddy currents induced by the moving magnets on a coaxial cylindrical conductor. The analysis underlined a damping effect that, for the analyzed system, grows with the cylinder thickness until it reaches 7 mm, above which the damping remains practically constant. Furthermore, the cylinder thickness, which reduces to one the number of overshoots in the moving magnet's position, grows with the clearance between the cylinder and the spring. Such an increase is not linear since it is 3 mm for a clearance of 0.5 mm, 4 mm for a gap of 1 mm, and 6 mm for a distance of 1.5 mm between the cylinder and the magnets.

Additional PMs can be introduced to obtain improved damping, and the whole device, also involving damping effects, can be considered for the optimization process.

**Author Contributions:** Conceptualization, N.G. and L.S.; methodology, A.M., R.R. and E.D.J.; software, A.M. and C.S.; validation, N.G., C.S. and L.S.; formal analysis, A.M. and E.D.J.; investigation, R.R. and N.G.; resources, L.S.; data curation, C.S.; writing—original draft preparation, A.M. and N.G.; writing—review and editing, N.G. and C.S.; visualization, C.S.; supervision, A.M., L.S. and E.D.J.; project administration, L.S.; funding acquisition, L.S. All authors have read and agreed to the published version of the manuscript.

**Funding:** Funder: Project funded under the National Recovery and Resilience Plan (NRRP), Mission 4 Component 2 Investment 1.3—Call for tender No. 1561 of 11.10.2022 of Ministero dell'Università e della Ricerca (MUR); funded by the European Union—NextGenerationEU. Award Number: Project code PE0000021, Concession Decree No. 1561 of 11.10.2022 adopted by Ministero dell'Università e della Ricerca (MUR), CUP I53C22001450006, Project title "Network 4 Energy Sustainable Transition—NEST.

**Conflicts of Interest:** The authors declare no conflict of interest.

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
