# Peer review of "Design and Optimization of a Permanent Magnet-Based Spring–Damper System"

_actuators, doi:10.3390/act12070291_

Round 1

Reviewer 1 Report

This paper reports the design process and the optimization of a permanent magnet-based spring. The paper research has reference value, but the following modifications are needed:

1) In the abstract "Computations were carried out by the research code EN4EM (Electric Network 4 ElectroMagnetics) developed by the authors.", Please give the corresponding calculation result, or please get the explanation conclusion after the calculation?

2) In the paper "Figure 5 reports the simulation results obtained with the commercial FEM package MagNet and EN4EM in terms of force-displacement characteristics.", Whether the optimization design is effective or not requires comparison. In this paper, the two kinds of software package are used for calculation and comparison, which seems not convincing enough, and most of it, the data before and after optimization is used for comparison.

3) In the conclusion section, only the qualitative expression results, we hope to use some quantitative indicators to express the conclusion.

No problem.

Author Response

The authors would like to thank the reviewer for his constructive comments that help in improving the quality of the manuscript. The responses to the reviewer’s comments are reported in italic just below the corresponding comment in the attached file.

Reviewer 2 Report

The manuscript reports the design process and the optimization of a permanent magnet-based spring. The cost function for the optimization procedure was built considering the difference between a required force-displacement curve and the force-displacement characteristic of a system composed of ring magnets axially polarized. For a given design, the cost function is evaluated using a semi-analytical model validated by FE simulations. A particular form of the Self-Organizing Map (SOM) algorithm was used to find the optimal configuration of the system able to give the desired force-displacement curve, resulting in an efficient design and optimization algorithm for permanent magnet-based springs.The logic of the manuscript is clear and innovative. However, for the sake of scientific rigor, there are some issues that need to be modified as follows.

1.     The manuscript lacks a clear and concise abstract that summarizes the main objectives, methods, and findings of the study. I suggest that the authors revise the abstract to provide a more comprehensive overview of the paper.

2.     Line 25 on page 1, high natural oscillation frequencies require that the mover mass (including a portion of the mass of the spring) is kept as low as possible, but longer strokes imply larger and heavier springs. Please refer to the relevant references to explain this change rule.

3.     In the introduction, the authors mention the need for a general methodology for evaluating the repulsive suspension force of a magnetic spring and analyzing the damping actions. Can you provide more background information on why this methodology is necessary and what specific applications or industries could benefit from it?

4.     Line 113 on page 3, It is known that in this arrangement of PMs, the repulsive force monotonically decreases as the clearance between the magnets increases and assumes its maximum when the clearance is zero. Please add references to explain the origin of this principle.

5.     Line 149 on page 4, Please explain the meaning of magnetostatic problem, what is the relationship between the magnetostatic problem and the eddy current problem mentioned below.

6.     Figure 3 on page 5, the upper right corner of the figure is not clear.

7.     Equation (2) on page 5, please explain the specific meaning of each letter in the equation.

8.     Figure 14 – Figure 19, 1.0mm, 1.5mm, 2.0mm, 3.0mm, 4.0mm, 5.5mm, 7.0mm, where does the choice of these numbers come from. A description should be given in the manuscript.

9.     In the methodology section, the authors mention the use of the EN4EM code for evaluating the suspension force and conducting electro-mechanical analysis. Can you provide more details on the capabilities and accuracy of this code? Has it been validated or compared against experimental data or other simulation tools?

10.  The manuscript mentions the optimization procedure for the magnetic spring, but it provides limited information on the specific optimization algorithm or approach used. Can you provide more details on the optimization method employed in this study? How was the objective function defined, and what constraints were considered?

English language and style need to be modified appropriately.

Author Response

(The authors gave the same response as above.)

Reviewer 3 Report

The article discusses the optimization of the force–displacement characteristic of a damping system based on a magnetic spring with a conductive damping tube. A manual optimization of the damping characteristics of this system was also carried out. The authors propose a semi-analytical model of the magnetic spring, verified using a commercial FEM software. The article is well written and easy to follow. Please address to the following comments:

1) Explain based on what the “Desired force–displacement characteristic” was obtained, shown in Figure 11.

2) A space is skipped in the caption to the figure “Figure11. Desired force–displacement characteristic.”

3) Why add the word “Aluminum” to the legend of every graph in Figures 14-19 if only aluminum tube is considered? Also, both the word “aluminum” and the word “aluminium” are used in the text. Please choose one correct form.

Author Response

(The authors gave the same response as above.)
